# Now it's your turn!: Eye blink rate in a Jenga task modulated by interaction of task wait times, effortful control, and internalizing behaviors

Kelley E. Gunther[1]*, Xiaoxue Fu[2], Leigha A. MacNeill[3], Morgan Jones[4], Briana Ermanni[5], Koraly Pérez-Edgar[6]

1 Department of Psychology, Yale University, New Haven, CT, United States of America, 2 Department of Psychology, The University of South Carolina, Columbia, SC, United States of America, 3 Northwestern University Feinberg School of Medicine, Chicago, IL, United States of America, 4 Department of Human Development and Quantitative Methodology, University of Maryland, College Park, MD, United States of America, 5 Department of Psychology, Virginia Tech, Blacksburg, VA, United States of America, 6 Department of Psychology, The Pennsylvania State University, State College, PA, United States of America

* kelleyegunther@gmail.com

**Data Availability Statement:** Data can be found at https://osf.io/q8jr9/

## Abstract

Dopamine is a versatile neurotransmitter with implications in many domains, including anxiety and effortful control. Where high levels of effortful control are often regarded as adaptive, other work suggests that high levels of effortful control may be a risk factor for anxiety. Dopamine signaling may be key in understanding these relations. Eye blink rate is a non-invasive proxy metric of midbrain dopamine activity. However, much work with eye blink rate has been constrained to screen-based tasks which lack in ecological validity. We tested whether changes in eye blink rate during a naturalistic effortful control task differ as a function of parent-reported effortful control and internalizing behaviors. Children played a Jenga-like game with an experimenter, but for each trial the experimenter took an increasingly long time to take their turn. Blinks-per-second were computed during each wait period. Multilevel modeling examined the relation between duration of wait period, effortful control, and internalizing behaviors on eye blink rate. We found a significant 3-way interaction between effortful control, internalizing behaviors, and duration of the wait period. Probing this interaction revealed that for children with low reported internalizing behaviors (-1 SD) and high reported effortful control (+1 SD), eye blink rate significantly decreased as they waited longer to take their turn. These findings index task-related changes in midbrain dopamine activity in relation to naturalistic task demands, and that these changes may vary as a function of individual differences in effortful control and internalizing behaviors. We discuss possible top-down mechanisms that may underlie these differences.

**Funding:** This material is based upon work supported by the National Science Foundation Graduate Research Fellowship under Grant No. DGE1255832 (to KEG) and by R21-MH111980 (to KPE). The funders had no role in study design, data collection and analysis, decision to publish, or preparation of the manuscript.

**Competing interests:** The authors have declared that no competing interests exist.

## Introduction

Proficient effortful control is typically linked with positive developmental outcomes. Effortful control includes abilities such as suppressing a dominant response in favor of a subdominant behavior and planning one's behaviors, contributing to broader profiles of self-regulation [1]. High levels of effortful control are frequently associated with behaviors tied to adaptive socioemotional development, including better emotion regulation [2], greater prosociality [3], and reduced risk of externalizing problems such as ADHD and impulsivity [3–7]. Effortful control is also frequently regarded as a protective mechanism against internalizing problems including anxiety and depression [4–6].

However, in considering adaptive levels of effortful control it may be possible to have "too much of a good thing." Specifically, high levels of effortful control may also be a risk factor for maladaptation in development. Murray and Kochanska [8] found an inverted "U" relation between level of effortful control and maternal report of internalizing, externalizing, and general problem behaviors in a longitudinal sample of children from toddlerhood to the preschool years. Moderate levels of effortful control predicted fewer problem behaviors, while higher incidence of problem behaviors were predicted by levels of effortful control at the two extremes—potentially reflecting profiles of over- and under-control. Additionally, the authors found that children with higher levels of effortful control had more parent-reported internalizing behaviors [8].

These perhaps counterintuitive relations between effortful control and adaptive development are also mirrored in the inhibitory control literature, as inhibitory control is a component part of the multifaceted construct of effortful control. Some work has found an inverse relation between inhibitory control and internalizing symptomatology in both children and adults [9–13]. However, other research has found that increased inhibitory control may actually serve as a risk factor for higher levels of internalizing behaviors [14, 15]. Overcontrolled behavior may potentiate attentional responses to environmental stimuli, particularly negatively valenced cues, thus prolonging periods of negative affect, which may pave the way for internalizing symptoms in development [16, 17].

### Dopamine signaling as a correlate of adaptive socioemotional behaviors and effortful control

There is limited research examining potential mechanisms underlying the idiosyncratic relations between effortful control and anxiety risk. Prior work, including from our own lab [18], suggests that dopaminergic activity may relate to individual differences in internalizing risk, as well as proficiency in behaviors encompassed by the effortful control umbrella.

Dopamine is a versatile neurotransmitter associated with a wide spectrum of social and nonsocial behaviors. Patterns of dopamine receptor binding generally show an inverse relation with anxiety symptomatology [19, 20]. In cross-species work, dopamine depletion is linked to anxiety-like behaviors [21]. For example, low counts of systemic dopamine D3 receptors were associated with increased anxiety-like behaviors in a mouse model [20]. Also, in humans, increased binding potential to D2 receptors in the medial prefrontal cortex (mPFC) and hippocampus was associated with decreases in reported social anxiety symptoms after treatment with cognitive behavioral therapy [19].

Dopaminergic activity is also associated with a host of regulatory behaviors encompassed by the broad term of effortful control. Dopamine D1 receptor binding has been associated with working memory, or the ability to retain information for the purposes of behavior-planning. These associations may also follow an inverted-U relation, where optimal levels of proficiency are actually found at average levels of dopamine/D1 receptor activation rather than at

the highest levels [22, 23]. Dopamine is also associated with inhibitory control proficiency, or the ability to withhold a dominant response in favor of a subdominant one [24 for review], and with attention shifting/cognitive flexibility, the ability to flexibly toggle between rule sets [25, 26]. Furthermore, dopamine is intimately linked to motivation and reward processes, playing a key role in approach and exploratory behaviors [27–29], which may moderate the implementation or proficiency of cognitive operations associated with effortful control [30].

However, dopaminergic activity is difficult to measure directly and non-invasively in human models because most common neuroimaging techniques do not reliably index neuro-chemical changes [31]. Much of what is known about associations between dopamine and behavior comes from animal work [e.g., 27, 32] or special populations with disorders charac-terized by low and high levels of dopamine, such as Parkinson's disease [e.g., 26, 33, 34] and Schizophrenia [e.g., 35], respectively. Techniques such as positron emission tomography (PET) or Transcranial direct-current stimulation (TcDS) that can track or stimulate dopami-nergic activity, respectively, are invasive, generally unforgiving to motion from the participant, and not often friendly for work in children [19, 25, 31, 36].

In humans, eye blink rate is regarded as a peripheral index of striatal dopamine activity [34, 37–39], specifically linked to striatal D1 and D2 receptors [38], which are in turn broadly related to both cognitive and emotional control [40]. Prior work finds that eye blink rate and dopamine activity are positively related, where increases in eye blink rate are associated with increases in dopamine binding [38, 39, 41], although the exact mechanism underlying this association remains unclear [42]. The eye blink to dopamine association has been validated through pharmacological studies in both animal [32, 41, 43] and human [38, 44] models, using dopamine agonists and antagonists, as well as in patient populations such as individuals with Parkinson's disease [33, 34, 45] or Schizophrenia [35, 38]. We do note, however, that research associating eye blink rate with striatal dopamine *synthesis* has been mixed [46] and to our best knowledge no work has tested the association between dopaminergic activity and eye blink rate in healthy children, in part due to ethical and methodological considerations.

Research with eye blink rate falls into two broad methodological categories: tonic eye blink rate and phasic eye blink rate. Studies using tonic eye blink rate measure eye blink rate during a longer baseline period and then associate eye blink rate with a separate behavioral measure, such as cognitive tasks [47]. In comparison, phasic eye blink rate studies look at task-related changes in eye blink rate, often using shorter time scales [47]. As early as infancy, eye blink rate may be rapidly modulated by current activity such as feeding or viewing different novel stimuli [48] and this sensitivity to stimuli continues through adulthood [49] suggesting the utility of each task design across all age ranges. Studies employing either phasic or tonic eye blink rate each contribute different, yet critical, information to the broader literature.

Prior work has found interrelations between tonic eye blink rate and inhibitory control. Zhang and colleagues [50] found in a sample of healthy adults that increased tonic eye blink rate was associated with increased accuracy as well as efficiency on a go/no-go task. However, directionality is not entirely consistent across this line of work. For example, Colzato and col-leagues [24] found that increased tonic eye blink rate was associated with *decreased* efficiency on a stop signal task in a sample of healthy adult participants. Looking to task-related changes in eye blink rate, Siegle and colleagues [51] found that eye blink rate increased as cognitive load increased on a Stroop task, also in a sample of healthy adults. As for attention shifting, another behavior related to effortful control, both Zhang and colleagues [50] and Tharp and colleagues [52] found that higher tonic eye blink rate was associated with better performance on an attention shifting task. However, there is minimal work investigating either of these associations in children.

Working memory, another behavior related to effortful control [53], has also been associated with eye blink rate. Zhang and colleagues [50] found that higher tonic eye blink rate was associated with lower proficiency on a working memory task in adults. Adding to this literature base, Ortega and colleagues [54] measured eye blink rate during wait periods of a working memory task and found that higher eye blink rate was associated with greater accuracy. Additionally, Rac-Lubashevsky and colleagues [55] found that phasic eye blink rate in adults changed with demands on a working memory task, with increased eye blink rate on trials that involved working memory updating and gate switching, which both require greater cognitive control from the participant. In infants, Bacher and colleagues [47] found that eye blink rate did not relate to performance on the classic A-not-B task assessing working memory, but did change as a function of task phase. Specifically, eye blink rate was significantly higher when the toy was hidden as compared to when it was revealed, suggesting that eye blink rate was higher for periods of the task with greater demands. Moreover, infants with greater fluctuation in eye blink rate between phases had higher accuracy on the working memory task.

Eye blink rate may also fluctuate according to other attentional processes, such as sustained attention, which are also mediated by dopaminergic processes [56]. As early as infancy, infants will blink less in response to moving stimuli designed to elicit sustained attention, as compared to baseline [57, 58]. Adults also display lower eye blink rates with higher sustained attention [59], suggesting continuity in this relation through development.

However, the data taken together reveal mixed findings in the nature of relations between eye blink rate and various cognitive processes. While higher tonic eye blink rate relates to more proficient attention shifting [50, 52], it also relates to less proficient working memory [50], and prior work has found both positive [50] and negative [24] relations between tonic eye blink rate and inhibitory control. Looking to work with task-related changes in eye blink rate, increases within an individual are commonly associated with increases in effort or task demands [47, 51, 55, 60]. Yet, findings pertaining to sustained attention [57–59] may contradict these findings, where effortful control may support sustained attention, and vice versa [61, 62]. Thus, additional work is needed to better understand these relations as well as individual differences in these associations.

Additionally, seemingly no published work has investigated direct relations between anxiety symptoms and eye blink rate. Barbato and colleagues [63] found a positive significant association between tonic eye blink rate and neuroticism, which they suggest may be a risk factor for anxiety disorders. Studies have also investigated associations between dopaminergic activity and behaviors that in part describe the anxious phenotype, as well as between eye blink rate and anxious behaviors. These behaviors include variation in emotion regulation, reward processing, reinforcement learning, and patterns of exploration versus exploitation [40, 64–67].

For example, in a sample of adolescents, Barkley-Levinson and Galván [68] found that tonic eye blink rate was positively associated with the tendency to maximize reward within a task, suggesting that, specifically for adolescents, dopamine binding may be positively related to reward sensitivity. Additionally, Van Slooten and colleagues [39] found that lower tonic eye blink rate was related to individuals exploiting familiar, higher-valued options in a task, while higher eye blink rate was associated with the propensity to explore unfamiliar, lower-valued options. This work maps onto prior findings, where exploitive tendencies are associated with anxiety and both anxiety and exploitive strategies are linked with lower dopamine binding [19, 20, 39].

In sum, prior work shows associations between multi-modal measurements of dopaminergic activity and effortful control, as well as with anxiety-related behaviors. However, only limited research utilizes eye blink rate, and many findings are inconsistent. Additionally, to date we are unaware of work that examines eye blink rate, anxiety, and effortful control within the same model to see how these constructs may interact.

## Naturalistic assessments of cognitive processes

Much of the reviewed work has relied on computer-based tasks, which allow for greater control and repetition. However, this control often comes at the cost of external validity. In addition, tasks seen in the adult literature are often not developmentally sensitive, leading to the use of alternate tasks. For example, many assessments of effortful control take the form of "games" that a child plays with the experimenter. In one such game, a snack delay task, a child must wait to retrieve a candy from a clear plastic cup. Alternately, children are asked to walk on a taped line as slowly as possible [2]. These kinds of paradigms offer ecological validity by more closely resembling encounters a child may have in real life and are more developmentally appropriate, versus a lab-based computer task [69, 70].

The Tower of Patience task has also been widely used to assess effortful control in children. The child and a familiar experimenter take turns either building a tower with blocks [3, 8, 71–75] or withdrawing blocks from a Jenga-style tower [76, 77]. With each turn, the experimenter follows a schedule of increasingly lengthened delays to take their turn, making the child wait longer to continue game play. Behavioral measures focus on different violations of the turn-taking rule, such as the child skipping the experimenter's turn and continuing to choose their own block, with the operationalization that less adherence to the turn-taking rule is associated with lower effortful control [8, 71, 73, 74].

While many tasks assessing facets of effortful control move beyond the computer screen, dopaminergic activity has traditionally been measured in more constrained settings. PET can be used to measure dopamine binding in specific brain areas, but requires minimal motion on the part of the participant and severely limits compatible tasks, as well as its use with children [19, 38]. Other work frequently uses electromyography or stationary eye tracking to quantify eye blink rates, but the associated tasks also require very little movement on the part of the participant and are also limited to a computer screen. Therefore, little is known regarding the relation between eye blink rate and behavior in more naturalistic paradigms, particularly when embedded in a social context.

Mobile eye tracking can capture an individual's ocular activity while ambulatory, via convenient setups worn by the participant [78]. An emerging literature has used mobile eye tracking to quantitatively measure visual attention patterns in naturalistic scenes in populations ranging from infancy to adulthood [79–81]. The same technology can be used to measure eye blinks while freeing the participant from the constraints of a computer screen [18]. Leveraging this technology, we can then capture eye blinks in-the-moment as children engage in a task eliciting effortful control.

## Current study

In this study, we collected phasic, event-related eye blink rate during the Tower of Patience game, in which children were asked to wait increasingly long periods of time to take their turn during a Jenga-like game. This task was designed to assess effortful control in a more true-to-life setting. We tested whether the duration that children were asked to wait for each trial, their parent-reported effortful control, their parent-reported internalizing symptoms, and/or the interaction of these variables, significantly related to eye blink rate, a peripheral measure of dopamine activity. Due to a paucity of prior research in this domain, our analyses were generally exploratory in nature. We did predict that eye blink rate would increase as trial wait time got longer and thus more challenging. However, we did not have hypotheses regarding how parent-reported internalizing symptoms and/or parent-reported effortful control would relate to these changes over the time course of the task.

## Method

### Participants

Participants in the current analyses were 55 children ranging from 5- to 7- years of age (M = 6.15 years, *SD* = 0.60, 49.1% female) identifying as White (87.3%), Asian (5.4%), African American (3.6%), Latino (1.8%), and other (1.8%), reflecting the demographics of the surrounding semi-rural community. Families were recruited using a University database of families expressing interest in participating in research studies, as well as community outreach and word-of-mouth. Children with high levels of Behavioral Inhibition (BI) were oversampled. BI is a risk factor for social anxiety disorder in childhood and adolescence [82] and the original study aims included examining naturalistic visual attention in the context of risk for anxiety. Seventeen children (30.91%) in the final analytic sample were classified as BI. Exclusion criteria for enrollment in the study included non-English speakers, gross developmental delays, or report of severe neurological or medical illnesses. All study procedures were approved by the Institutional Review Board at the Pennsylvania State University. All parents and children completed written consent/assent and were compensated for their time.

To reach the analytic sample of 55 children, 163 children were first screened for BI via parent report with the Behavioral Inhibition Questionnaire [BIQ; 83]. Consistent with the previous literature [84–86], children were recruited as a BI participant if their total BIQ score was greater than or equal to 119 or if their social novelty subscale score was greater than or equal to 60. Of the full screening sample, 39 children (23.9%) met the BI criteria.

After screening, 70 children (20 BI) were brought to the lab to complete a battery of episodes assessing temperamental reactivity, including the "Tower of Patience" episode included in these analyses (described further below). The mean age of the sample was 6.11 years (*SD* = 0.60) with 34 females (48.8%). The sample predominantly identified as White (*n* = 61, 87.1%). Participants were excluded from the final analyses due to: technical problems (*n* = 9), requesting removal of the eye tracker (*n* = 1), non-completion of the game (*n* = 4), and being the twin of another participant (*n* = 1). Fig 1 depicts a visualization of participant recruitment.

### Behavioral inhibition

Parents completed a series of online questionnaires about themselves and their children prior to the laboratory visit. The BIQ [83] includes 30 questions that assess a child's response to novelty, using a likert scale ranging from 1 ("Hardly Ever") to 7 ("Almost Always"). While the BIQ was used to recruit participants and enrich the sample categorically, BI was assessed as a continuous variable in the analyses, such that higher scores reflected higher levels of BI (*M* = 92.82, *SD* = 27.48). The BIQ had excellent internal consistency in this study (Chronbach's α = 0.95).

### Effortful control

Effortful control was measured via parental report with the very short form of the Children's Behavior Questionnaire [CBQ-VSF; 87]. The CBQ-VSF includes 36 questions that assess aspects of a child's temperament including surgency, negative affect, and effortful control. A 7-point likert scale is used representing responses ranging from "extremely untrue of your child" to "extremely true of your child." We used the effortful control subscore as a continuous variable in our analysis, which is related to inhibitory control, attentional control, low intensity pleasure, and perceptual sensitivity [88]. Higher scores on this subscale reflect higher levels of effortful control (*M* = 5.10, *SD* = 0.67). The CBQ-VSF had good internal consistency in this study (Chronbach's α = 0.69).

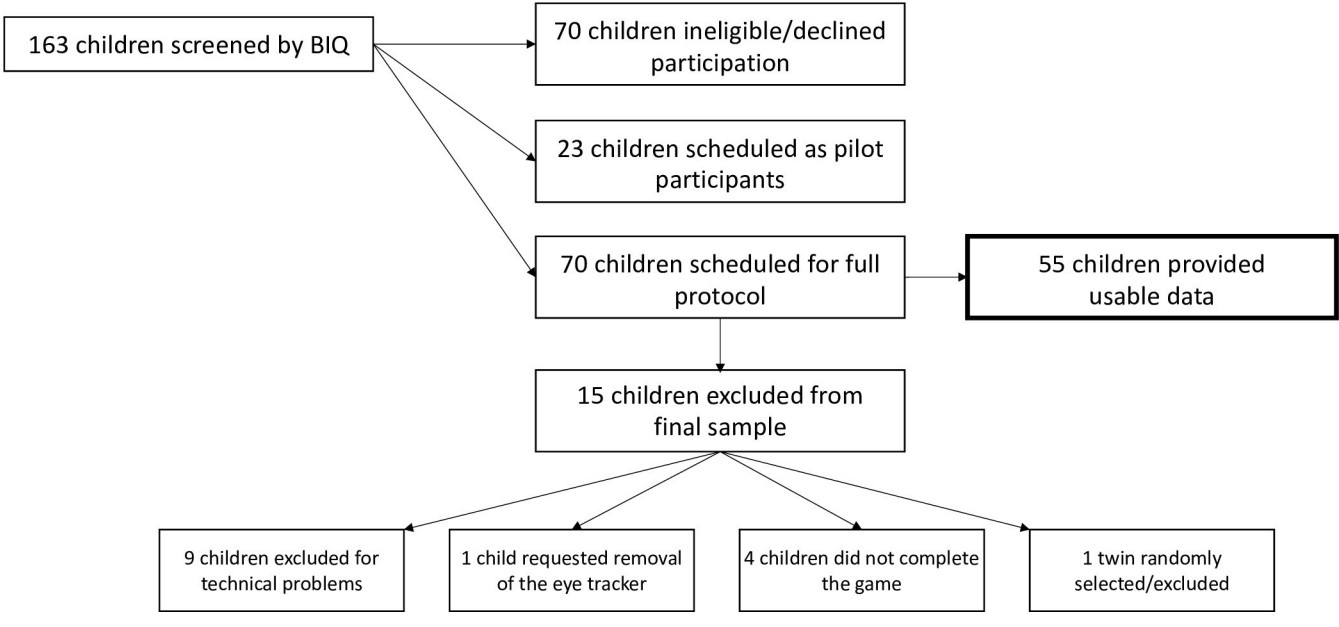

**Fig 1. Visualization of participant recruitment.**

## Internalizing behaviors

Internalizing behaviors were measured via the internalizing subscale of the Child Behavior Checklist [CBCL; 89] and assessed as a continuous variable, such that higher values reflected a higher count of reported internalizing behaviors. We chose to use count of internalizing behaviors in these analyses rather than count of anxiety symptoms. While anxiety disorders are seen in children as young as preschool-age [90], it is more common for onset to be as late as adolescence or adulthood [91]. Because the sample for the current study is both young and healthy, there is relatively low likelihood that many children in this sample will display symptoms at or near a clinical threshold. Measuring internalizing behaviors is a broader classification that includes symptoms of anxiety, thus offering greater analytic variability and a better developmental match within the sample.

The mean count of internalizing behaviors in this sample was 6.09 ($SD$ = 5.10, $range$ = 0–25). The internalizing subscale of the CBCL also had good internal consistency in this study (Chronbach's α = 0.84).

## Tower of Patience task

The Tower or Patience task was used to elicit effortful control. In this episode, the child was seated at a table and introduced to a Jenga-like game, where they took turns playing with a familiar experimenter. They were told that they would alternate withdrawing wooden blocks from a vertically stacked tower and needed to avoid the tower's collapse. Each selected block was placed in an adjacent box after every turn. The blocks were three different colors, with each vertical third of the tower colored either blue, yellow, or red. Based on these colors the child was also introduced to a scoring scheme, where the lower third of blocks in the tower were worth three points if selected, the middle third were worth two points, and the final top third were each worth one point. The children were told that the player with the most points at the end of the game won. With each subsequent turn, the experimenters increasingly delayed in choosing a block to remove from the tower. The increasing delays were presumed to be

increasingly frustrating for the child, and thus required greater effortful control to adhere to the turn-taking rules.

While a "naturalistic" task, the task was constrained in that the experimenter was provided time intervals to adhere to as closely as possible during the game. There were 7 trials with time intervals as follows: trial 1 = no wait period, trial 2 = 10 second wait period, trial 3 = 20 second wait period, trial 4 = 30 second wait period, trial 5 = no wait period, trial 6 = 40 second wait period, trial 7 = 60 second wait period. After the final 60-second wait period, the experimenter would "accidentally" knock down the tower to end game play.

If the tower accidentally fell over during the progression of trials, the experimenter would re-build the tower, take a turn with no wait period, and then continue with the next specified trial. During each wait period, the experimenter was instructed to keep gaze and behavior ambiguous, so it was not clear to the child what was causing the delay. If the child spoke to the experimenter during the wait period, the experimenter either disregarded the child or provided a brief non-committal answer. If the child violated the turn-taking rule, the experimenter would wait until the trial was over to remind the child, "Remember how to play this game. First, I take a block, and then you take a block, then I take one, then you take one. That's how we play this game." Any subsequent violations of turn-taking were left unacknowledged. Four research assistants acted as the primary experimenter in the current sample (all female).

A coding scheme was developed to mark the onset and offset of each wait period, as well as the child's violations of the turn taking rule and their verbal and physical prompts to the experimenter to take their turn. Behavior was coded using Datavyu [92]. A combination of the child's mobile eye tracking footage, a video captured of the scene using a video camera set up behind a two-way mirror, and audio recorded in the room was used to provide the most comprehensive coverage of behavior during the episode.

Trial onset was defined by when the child's selected block hit the bottom of their box used to store the drawn blocks. The trial offset was marked when the experimenter's selected block hit the bottom of their box, and/or the experimenter verbally told the child, "Now it's your turn!" If trials were out of order due to experimenter error or due to the tower falling before the game was completed, the coders adjusted the label of the trial such that it most closely matched the amount of time that the child had to wait, rather than the temporal order of trials originally established in the protocol.

A violation of turn taking was defined as the child removing blocks from the tower before the experimenter had chosen their block. The onset of the violation of turn taking was defined as the child touching the block in the tower. A verbal prompt was defined as the child encouraging the experimenter to take their turn or commenting on how long they had been waiting. Examples include, "It's your turn" or "Why do you take so long?" Chatter related to the game but not pertaining to the wait time or the experimenter's pending turn (e.g., "I used to play this game at home.") was not coded as a verbal prompt. A physical prompt was defined as physically directing the experimenter's attention to the tower, which included pushing a block toward the experimenter, pointing at the tower, or attempting to select but not actually withdrawing a block. Gestures accompanied by vocalizations pertaining to the child's own self planning (e.g., "What if I pick this one?") were not coded as a physical prompt.

Summary variables computed included the total number of verbal prompts, physical prompts, and turn skips across all wait periods, as well as the latency to the child's first verbal prompt. In coding the latency, a wait period was scored as the full time as per the protocol if the child makes no verbal prompts. If they did make a prompt, the onset of that wait period during which the prompt occurred was subtracted from the onset of the verbal prompt and rounded to the nearest second. This value and the duration of all previous wait periods were summed. For this computation, any wait period durations that exceeded the prescribed wait time were truncated to the

maximum wait time for that trial. If the child made no verbal prompts through the episode, they were then assigned the total value of all wait periods in the protocol, 160 seconds.

Three independent coders completed behavioral coding for the sample, overlapping on 24% of videos to ensure reliability. Frame-by-frame reliability for trial onsets/offsets as well as coded behaviors was computed across coders. Coders agreed on 94% of frames denoting the onset and offset of each trial, 98.3% of frames in which there was a verbal prompt, 99.3% of frames in which there was a physical prompt, and 99.5% of frames in which there was a turn skip.

In order to minimize noise in our naturalistic paradigm, data were cleaned on a trial-by-trial basis to retain trials that adhered as closely as possible to the protocol. Trials in which the trial duration was more than 2 standard deviations above or below the mean trial duration were removed. This resulted in the removal of 17 trials in total (out of 269). Additionally, trials that were skipped by the experimenter in error (N = 1) were treated as missing data. Means and standard deviations of the cleaned trial durations can be seen in Table 1.

## Ambulatory eye tracker

Participants wore a Pupil binocular ambulatory eye tracker [Pupil Labs; 93] to record their eye blinks throughout the Tower of Patience task. The headset consists of two separate cameras, each pointing at an eye, as well as a camera centered on the space immediately in front of the child, capturing their world view. Data were recorded either with Pupil Capture v.0.9.6 (Pupil Labs) installed on a Microsoft Surface Pro 3 tablet with Windows 10 used in an earlier phase of the larger study (n = 12 in final sample) or with Pupil Capture v.0.9.12 (Pupil Labs) Installed on a MSI VR One Backpack PC also running Windows 10 (n = 43 in final sample). A monitor located in a separate room was remotely connected to the PC enclosed within the backpack for real-time monitoring of data quality during the experiment. The headset plus the backpack were light enough so as not to hinder naturalistic movement during the session. Data collection occurred in a room with no windows, so ambient light was consistent across all participants.

Eye blinks were event coded during each wait period of the Tower of Patience task using Datavyu [92]. Two independent coders completed behavioral coding for the sample, overlapping on 33% of videos to ensure reliability. Videos were a resolution of 1920x1080 pixels and a frame rate of 30 frames per second. To be considered a blink, both eyes had to close. Sustained closures of the eyelid (i.e., eyes completely closed for more than 1 frame) were not coded as blinks. Reliability between coders was calculated using a paired sample $t$-test, showing statistically comparable codes ($t = 0.89$, $p = 0.38$). Descriptive statistics for eye blink rate per trial can be seen in Table 2.

Eye blink rate for each wait period was computed by dividing the number of coded blinks within each wait period by the coded duration of the wait period in seconds, yielding a unit of blinks per second.

**Table 1. Mean, standard deviation, and range of the duration of each task trial in seconds.** Trials 1 and 5 are not listed as they did not tax inhibitory control nor have a standardized wait time.

| Trial Number | Mean duration (seconds) | Standard Deviation (seconds) | Range (seconds) |
|:---:|:---:|:---:|:---:|
| Trial 2 | 15.95 | 3.74 | 7.32–25.15 |
| Trial 3 | 26.48 | 4.02 | 17.36–36.04 |
| Trial 4 | 37.69 | 4.04 | 30.06–46.79 |
| Trial 6 | 49.43 | 5.95 | 36.36–63.70 |
| Trial 7 | 64.00 | 7.50 | 45.45–83.64 |

Note: Trials 1 and 5 did not require the participant to wait to take their turn

**Table 2. Descriptive statistics for eye blink rate for each task trial.** Trials 1 and 5 are not listed as they did not tax inhibitory control nor have a standardized wait time.

| Trial Number | Mean EBR | Standard Deviation | Range | Skew | Kurtosis |
|---|---|---|---|---|---|
| Trial 2 | 0.11 | 0.1 | 0–0.41 | 0.69 | -0.16 |
| Trial 3 | 0.12 | 0.12 | 0–0.50 | 1.45 | 1.32 |
| Trial 4 | 0.12 | 0.11 | 0–0.49 | 1.24 | 1.15 |
| Trial 6 | 0.10 | 0.08 | 0–0.35 | 1.28 | 1.32 |
| Trial 7 | 0.11 | 0.10 | 0–0.35 | 1.19 | 0.23 |

Note: Trials 1 and 5 did not require the participant to wait to take their turn

## Data analysis

**Descriptive statistics.** With the relative novelty of this experimental design, we first sought to describe these behavioral and physiological measures within our sample and how they may correlate.

**Multilevel model.** We used multilevel modeling to examine the interaction between level of effortful control, level of internalizing symptoms, and the length of trial in seconds on repeated measures of eyeblink rate. BIQ score was entered as a covariate in modeling to account for our original sampling scheme, where children were specifically recruited for elevated reported levels of BI. Length of trial was also entered as a random effect. All variables in the model were continuous.

## Results

### Descriptive statistics

Associations between behavioral and physiological measures within our sample as well as descriptive statistics for these variables can be found in Tables 3 and 4.

### Multilevel model

Our multilevel model was based on 252 repeated measures of eye blinks per second, nested within 55 persons. The results from the multilevel model can be seen in Table 5.

Of note, there was a significant three-way interaction between internalizing symptoms, effortful control, and trial wait time on eye blink rate, $b < 0.001$, $p = .04$. There was also a trend-level main effect of trial wait time on eye blink rate, $b = 0.01$, $p = .06$.

To further understand the nature of this three-way interaction, we probed the interaction with simple slopes testing. Both internalizing behaviors and levels of effortful control were split into low, medium, and high levels by grouping at -1 SD, mean, and +1 SD, respectively.

**Table 3. Descriptive statistics for behavioral and questionnaire-based variables.**

| Measure | *M* | *SD* | Range | Skew | Kurtosis |
|---|---|---|---|---|---|
| Latency to first turn skip (seconds) | 104.3 | 74.10 | 0.00–160.00 | -0.61 | -1.59 |
| Latency to first verbal prompt (seconds) | 35.24 | 49.21 | 0.00–160.00 | 1.28 | 0.19 |
| Number of verbal prompts | 2.76 | 3.46 | 0–14 | 1.53 | 1.66 |
| Number of physical prompts | 0.82 | 1.43 | 0–6 | 1.85 | 2.79 |
| Number of turn skips | 0.33 | 1.23 | 0–6 | 3.74 | 12.65 |
| Internalizing symptoms (CBCL) | 6.09 | 5.10 | 0–25 | 1.55 | 3.45 |
| Effortful control (CBQ) | 5.10 | 0.67 | 3.00–6.33 | -0.56 | 0.34 |
| BIQ | 92.82 | 27.48 | 43–149 | -0.05 | -0.98 |

**Table 4. Spearman's correlation table showing interrelations between demographic variables and coded behavioral variables.** Age and sex were included in correlations to test for any significant differences and inform subsequent model. Numbers 1 through 9 on the horizontal axis align with variables 1 through 9 on the vertical axis.

| | 1 | 2 | 3 | 4 | 5 | 6 | 7 | 8 | 9 |
|---|---|---|---|---|---|---|---|---|---|
| 1. Latency to first turn skip (seconds) | - | | | | | | | | |
| 2. Latency to first verbal prompt (seconds) | 0.74*** | - | | | | | | | |
| 3. Number of verbal prompts | 0.73*** | 0.36** | - | | | | | | |
| 4. Number of physical prompts | 0.44*** | 0.15 | 0.66*** | - | | | | | |
| 5. Number of turn skips | -0.21 | 0.13 | 0.10 | 0.02 | - | | | | |
| 6. Internalizing symptoms (CBCL) | -0.01 | 0.06 | -0.05 | -0.16 | 0.04 | - | | | |
| 7. Effortful control (CBQ) | -0.03 | 0.07 | -0.05 | 0.02 | 0.15 | -0.04 | - | | |
| 6. Sex | -0.15 | -0.19 | -0.06 | -0.08 | -0.06 | -0.09 | 0.32* | - | |
| 9. Age (years) | -0.16 | -0.09 | -0.26+ | -0.07 | -0.04 | -0.19 | -0.16 | -0.15 | - |
| 10. BIQ | -0.12 | -0.08 | -0.21 | -0.15 | -0.16 | 0.56*** | 0.01 | 0.00 | 0.05 |

$+p < .1$,
$*p < .05$,
$**p < .01$,
$***p < .001$. Sex: 0 = male, 1 = female

Here we found that for children with low (-1 SD) internalizing and high (+1 SD) effortful control, blinks per second significantly decreased as trial wait time increased, $b < -0.001$, $p = .02$ (Fig 2). For all other levels of internalizing behaviors and effortful control, the relation between trial wait time and eye blink rate was not significant.

We noted from descriptive statistics that child sex was significantly related to reported effortful control and that child age was related to number of verbal prompts at trend level. We

**Table 5. Results from multilevel model showing experimental variables moderating within-person differences in eye blink rate per trial of task.**

| | Est. | SE | t |
|---|---|---|---|
| Fixed effects | | | |
| Intercept | -0.08 | 0.20 | -0.39 |
| BI | < -0.001 | < 0.001 | -0.58 |
| Internalizing symptoms | 0.03 | 0.02 | 1.34 |
| Effortful control | 0.04 | 0.04 | 1.11 |
| Trial wait time | 0.01+ | < 0.01 | 1.87 |
| Internalizing * Eff. control | -0.01 | < 0.01 | -1.24 |
| Internalizing * Trial wait time | < -0.001* | < 0.001 | -2.11 |
| Eff. control * Trial wait time | < -0.01* | < 0.001 | -2.01 |
| Internalizing * Eff. control * Trial wait time | < 0.001* | < 0.001 | 2.11 |
| Random effects | | | |
| Trial wait time | < 0.001 | | |
| Residual | 0.05 | | |

*Note*: Model based on 252 repeated measures of eye blinks per second, nested within 55 persons.
$+p < .10$,
$*p < .05$,
$**p < .01$,
$***p < .001$

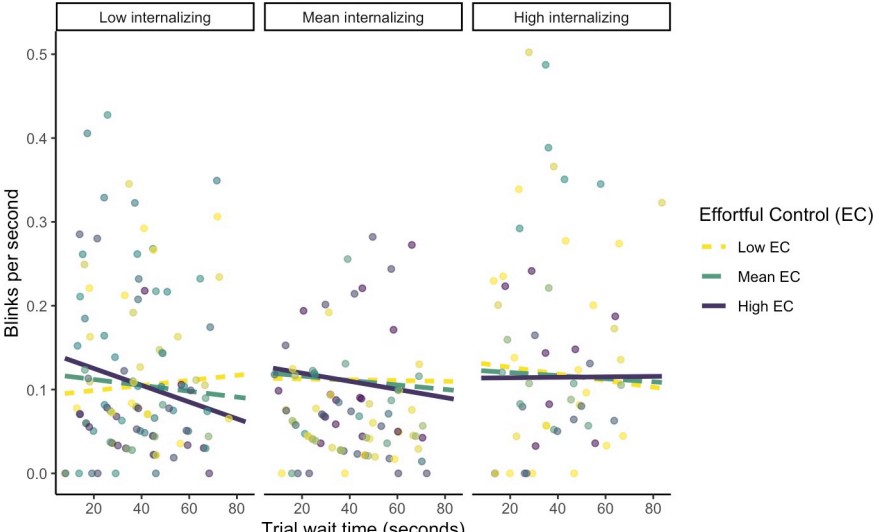

**Fig 2. Graph probing three-way interaction between trial wait time, effortful control, and internalizing symptoms on blink rate.** BI is included as a covariate in this analysis.

tested an additional model entering these variables as additional covariates and all effects reported above were the same in significance and directionality.

## Discussion

Prior work suggests that dopamine has far-reaching correlates in many domains of development, including core cognitive and socioemotional processes. Further understanding dopamine neurotransmission as it pertains to childhood behavior may help to elucidate the mechanisms that link effortful control and risk for social maladaptation, like anxiety disorders. However, measuring dopaminergic activity directly is logistically difficult and paradigms are largely limited to PET scans or pharmaceutical manipulations. These constraints have historically limited both investigations involving healthy children, as well as the type of paradigms that can be used while dopaminergic activity is concurrently measured.

Eye blink rate provides a peripheral measure of midbrain dopaminergic activity and advances in ambulatory data collection have made it feasible to measure eye blink rate during a naturalistic task in real time [18]. Here, we recorded eye blinks while children participated in the Tower of Patience episode, a task designed to elicit and assess effortful control. Wait periods for the child to take their turn got increasingly longer as the task progressed. Thus, with successive trials the task became more difficult on two different dimensions: first, the child was asked to wait longer, and second, the game had been ongoing for a longer time and the child was dealing with an accumulation of these long wait periods.

Notably, participants provided a relatively low level of analytic variability in their behavior. Indeed, only 3 children outright skipped the experimenter's turn when the task requested that they wait until the experimenter chose a block. Additionally, 31% of the sample (n = 17) did not make any verbal prompts during these wait periods. Finally, these overt behaviors were not significantly correlated to effortful control or internalizing behaviors as assessed by parent report (Tables 3 and 4). This relative homogeneity in overt behavior coupled with underlying variation in attentional and neural processes is not uncommon in developmental work. In an overlapping sample, the research group found few behavioral differences as children interacted with a stranger wearing a scary gorilla mask, but did find that level of BI predicted proportion

of gaze allocated to the stranger's head/mask (the potential source of threat) over time [94]. Similarly, Wolfe and Bell [13] found no behavioral differences in executive functioning ability as a function of shyness in a sample of preschoolers, but found that medial frontal EEG power differentiated between these groups.

This relative homogeneity in behavior provided a unique opportunity to investigate how changes in dopaminergic activity over the course of task demands may relate to or characterize individuals within the sample, above and beyond overt behaviors. While children across the sample had similar net behaviors, different neural mechanisms may have supported this response for different children. In this analysis we found a significant three-way interaction between reported effortful control, reported internalizing symptoms, and task wait duration on eye blink rate. Probing the interaction revealed that for children with low reported internalizing symptoms and high reported effortful control, their eye blink rate significantly decreased as wait times during the game increased. Therefore, for children with an effortful control and internalizing profile that is typically be considered adaptive, we could propose that midbrain dopamine binding decreased as the task became more taxing for the child. We predicted that eye blink rate would increase as trials got longer and thus more difficult, in contrast to the pattern that emerged.

We note that hypothesis-building was difficult due to the relative novelty of this research and the mixed existent findings. Generally, decreased dopamine levels are associated with greater anxiety, and published associations between dopamine and attention remain mixed. Where some work has found a positive relation between tonic eye blink rate and efficiency on cognitive tasks calling upon attention shifting [50, 52], work with working memory finds an inverse relation between tonic eye blink rate and working memory proficiency [50]. Work with inhibitory control also remains mixed, with some authors finding a positive relation between tonic eye blink rate and inhibitory control performance [50], while other authors report an inverse relation [24]. Looking to changes in eye blink rate within a task, the literature generally finds that increases in task demands relate to increases in eye blink rate [47, 51, 55]. However, running parallel and perhaps in contradiction to these findings [61, 62], separate work finds that decreases in eye blink rate are associated with increased sustained attention [57–59]. Within the context of this experiment, we cannot determine which mechanism or combination of mechanisms may underlie these significant task-related changes, or lack thereof, in eye blink rate. Therefore, future research should more directly investigate more specific cognitive mechanisms of changes in phasic eye blink rate.

We posit that these task-related decreases in eye blink rate, specifically for children with relatively high effortful control and relatively low internalizing symptoms, may reflect some combination of the following mechanisms. First, where increased phasic eye blink rate has been attributed to increased cognitive load or task demands [47, 51, 55], it may be the case that these children may become more efficient in inhibiting a prepotent response to skip the experimenter's turn as the task progresses. Alternately, where eye blink rate is positively associated with reward value and sensitivity [68, 95], decreases in eye blink rate may also reflect either conscious or subconscious changes in reward valuation for these children as the task becomes increasingly taxing. Another explanation is that as the trials get more taxing, both in terms of wait time and the instability of the tower/the risk of the tower falling, task-related decreases in eye blink rate may be reflecting increases in sustained attention for these children [57–59]. Additional work will be needed to disentangle these competing processes.

This study is not without its limitations. We acknowledge a modest sample size in our analyses, although utilizing repeated measures of eye blink rate provided the necessary power to conduct these analyses. Indeed, our multilevel model included 252 observations. Additionally, while our task had increased ecological validity compared to the majority of prior work examining dopamine neurotransmission, our sample lacked generalizability due to

sociodemographic homogeneity. Additionally, we saw these data as an opportunity to investigate dopaminergic activity as a way to describe children with otherwise relatively invariant behaviors in response to the task demands. However, future research would benefit from analyses assessing interrelations between eye blink and overt behavior on similar tasks, and how eye blink rate may relate to rule violations during an effortful control task. This could be accomplished by administering a more demanding task, for example.

Finally, we recognize that direct associations between eye blink rate and dopamine have emerged from a predominantly clinical literature [e.g., 26, 33–35] as well as more invasive adult tasks [e.g., 19, 25, 31, 36]. As such, limited work has examined associations between dopamine and eye blink rate in healthy children. Therefore, it may be the case that changes in eye blink rate are related to attentional processes more broadly [47, 51, 55, 57–59] and more distally related to dopaminergic activity itself.

Additionally, we note that our study did not include an analysis of tonic eye blink rate. The main focus of the original study was eye gaze rather than eye blink rate, but we recognized the flexibility of mobile eye tracking in the acquisition of additional measures. Therefore, baseline eye blink rate was not included in the protocol. However, in work assessing task-related changes in eye blink rate, the absence of a baseline period is not unusual (e.g., [47]). Moreover, with children of this age, it is difficult to acquire a period of data in which the child remains relatively still and neutral. Finally, due to the task- and context-sensitivity of eye blink rate, and how eye blink rate may change as a function of small environmental changes, the labeling of a baseline period post-hoc was deemed too noisy and thus not extracted for analysis [49].

Taken together, these findings provide methodological proof of concept in measuring task-related changes in eye blink rate during a naturalistic task. We also find that changes in eye blink, measuring fluctuations in dopaminergic activity during changes in task demands, may vary as a function of effortful control and internalizing symptoms. These findings present dopamine binding as another important factor to consider in better understanding individual differences in both cognitive and socioemotional development.

## Author Contributions

**Conceptualization:** Kelley E. Gunther, Xiaoxue Fu, Koraly Pérez-Edgar.

**Data curation:** Kelley E. Gunther.

**Formal analysis:** Kelley E. Gunther.

**Investigation:** Kelley E. Gunther, Xiaoxue Fu, Leigha A. MacNeill, Morgan Jones.

**Methodology:** Kelley E. Gunther, Xiaoxue Fu, Leigha A. MacNeill.

**Project administration:** Xiaoxue Fu, Leigha A. MacNeill, Briana Ermanni.

**Supervision:** Koraly Pérez-Edgar.

**Writing – original draft:** Kelley E. Gunther.

**Writing – review & editing:** Kelley E. Gunther, Xiaoxue Fu, Leigha A. MacNeill, Morgan Jones, Briana Ermanni, Koraly Pérez-Edgar.

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
