## [Decision Letter · Decision Letter 0]

5 Jul 2023

PONE-D-22-35097Now it’s your turn!: Eye blink rate in a Jenga task modulated by interaction of task wait times, effortful control, and internalizing behaviorsPLOS ONE

Dear Dr. Gunther,

Thank you for submitting your manuscript to PLOS ONE. After careful consideration, we feel that it has merit but does not fully meet PLOS ONE’s publication criteria as it currently stands. Therefore, we invite you to submit a revised version of the manuscript that addresses the points raised during the review process.

We look forward to receiving your revised manuscript.

Kind regards,

Pierre Pouget

Academic Editor

PLOS ONE

Journal Requirements:

Reviewers' comments:

Reviewer's Responses to Questions

**Comments to the Author**

1. Is the manuscript technically sound, and do the data support the conclusions?

Reviewer #1: Yes

Reviewer #2: Partly

2. Has the statistical analysis been performed appropriately and rigorously? 

Reviewer #1: Yes

Reviewer #2: Yes

3. Have the authors made all data underlying the findings in their manuscript fully available?

Reviewer #1: Yes

Reviewer #2: Yes

4. Is the manuscript presented in an intelligible fashion and written in standard English?

Reviewer #1: Yes

Reviewer #2: Yes

5. Review Comments to the Author

Reviewer #1: 1. More information is needed in Methods regarding how blinks were measured. What was the sampling rate for measuring the blinks? It is stated that the camera tracked both eyes. Which eye was used to measure blinks? Dominant, left, right, or both?

2. Are the participants holding anything relevant in working memory for performance during this task? The literature review on blinks and working memory is outdated. More has been done since Zhang's 2015 paper. Recent phasic work indicates that spontaneous blinks during the working memory delay period correlate in a positive relation with performance. An updated literature search and Discussion is recommended especially with respect to the wait times that were used in this study (up to 60 seconds) and previous studies.

3. It is arguable that wait times longer than 30 seconds are beyond classically defined working memory. Does the 3 way interaction tell us anything useful for blinks during wait times that encompass classical working memory (up to 30 seconds) versus wait times that would classically be considered beyond working memory (40 and 60 seconds). This issue should also be mentioned in an expanded Discussion of blinking, working memory, and the delay (or maintenance period).

Reviewer #2: This article uses eye blink rate (EBR) as a proxy to evaluate dopamine neurotransmission in healthy children during an “ecologic” behavioural Jenga task. In this task, the measurement of EBR appears to be a feasible and interesting way to search for correlations between behaviour and physiology.

Major issues:

This article (introduction and discussion) should emphasize more that the link between dopamine and EBR has been established in patients (Parkinson disease and schizophrenia mostly), and in pharmacological studies. This link, to my knowledge, is not clear in healthy adults and recent data from van den Bosch et al. found evidence for absence of links between striatal dopamine synthesis capacity and spontaneous eye-blink rate. Moreover, it seems to me that this link has never been established in children.

Could you provide proofs from the literature of the link between EBR and dopamine in healthy children? In absence of evidence, and this is my main concern about the conclusions of this study, spontaneous eye blink rate of healthy children can’t be used as a proxy of dopamine synthesis in absence of scientific evidence proving the relation between dopamine neurotransmission and EBR in this specific population. The discussion about dopamine should then be more nuanced. The last sentence of the conclusion should be revised. The use of “dopamine” as a key word for this study is then questionable.

Minor issues:

In the introduction, authors should define more the terms “externalizing problems” and “internalizing problems”.

In introduction 2nd paragraph, “additionally, the authors found … behaviours”: is this sentence a result of a study (then a quote is needed), or more a general observation/opinion?

In Methods, concerning code of blink: the threshold used to determine between blink and sustained eye closure should be indicated.

First sentence of “descriptive statistics” “With the relative novelty of this experimental design, I first sought to describe these behavioral and physiological measures within our sample and how they may correlate.” should be revised.

The absence of baseline could be problematic, as you mention in discussion. In the trials 1 and 5, what was the mean duration of task execution? Is the EBR during those short periods an available data?

Concerning results, they should be more detailed in the text part.

On table 4, what the 1 to 8 stand for?

In discussion, could you comment on the fact that with successive trials, the game itself may become more difficult with maybe a more unstable tower and more at risk of falling, that may influence the attention onto the game by itself.

In discussion, the sentence “Looking to changes in EBR within a task, work generally finds that increases in task demands relate to increases in task demands” needs to be rephrased.

6. PLOS authors have the option to publish the peer review history of their article (what does this mean?). If published, this will include your full peer review and any attached files.

Reviewer #1: No

Reviewer #2: **Yes: **Quentin Salardaine

---

## [Author Response · Author response to Decision Letter 0]

19 Sep 2023

Reviewer #1: 

1. More information is needed in Methods regarding how blinks were measured. What was the sampling rate for measuring the blinks? It is stated that the camera tracked both eyes. Which eye was used to measure blinks? Dominant, left, right, or both?

We thank the reviewer for these questions. We added to page 20, “Videos were a resolution of 1920x1080 pixels and a frame rate of 30 frames per second. To be considered a blink, both eyes had to close.”

2. Are the participants holding anything relevant in working memory for performance during this task? The literature review on blinks and working memory is outdated. More has been done since Zhang's 2015 paper. Recent phasic work indicates that spontaneous blinks during the working memory delay period correlate in a positive relation with performance. An updated literature search and Discussion is recommended especially with respect to the wait times that were used in this study (up to 60 seconds) and previous studies.

We apologize for any confusion in the manuscript - the Jenga task is not meant to target working memory, but rather effortful control broadly. Our literature review touched upon associations between eye blink rate and working memory (as well as attention shifting and inhibitory control) only to tie eye blink rate to attentional/regulatory processes broadly construed. 

We have updated the introduction to include some more recent publications with eye blink rate and behaviors related to effortful control including Magliacano et al., 2020 and Ortega et al., 2022. 

3. It is arguable that wait times longer than 30 seconds are beyond classically defined working memory. Does the 3 way interaction tell us anything useful for blinks during wait times that encompass classical working memory (up to 30 seconds) versus wait times that would classically be considered beyond working memory (40 and 60 seconds). This issue should also be mentioned in an expanded Discussion of blinking, working memory, and the delay (or maintenance period).

Again, we apologize for the confusion. The Jenga task was primarily measuring effortful control and building on the prior literature deploying the game as a test of inhibitory control (Buss et al., 2014; Durbin et al., 2007; Dyson et al., 2012; Kochanska et al., 1996; Murray & Kochanska, 2002; Pereira et al., 2021; Ruf et al., 2008; Smith et al., 2013; von Suchodoletz et al., 2009). 

Reviewer #2: This article uses eye blink rate (EBR) as a proxy to evaluate dopamine neurotransmission in healthy children during an “ecologic” behavioural Jenga task. In this task, the measurement of EBR appears to be a feasible and interesting way to search for correlations between behaviour and physiology.

We thank the reviewer for their helpful and thoughtful comments. 

Major issues:

1. This article (introduction and discussion) should emphasize more that the link between dopamine and EBR has been established in patients (Parkinson disease and schizophrenia mostly), and in pharmacological studies. This link, to my knowledge, is not clear in healthy adults and recent data from van den Bosch et al. found evidence for absence of links between striatal dopamine synthesis capacity and spontaneous eye-blink rate. Moreover, it seems to me that this link has never been established in children.

We thank the reviewer for this important point. We have elaborated on page 6 as follows, “In humans, eye blink rate is regarded as a peripheral index of striatal dopamine activity (Eckstein et al., 2017; Jonkees & Colzato, 2016; Karson, 1983; Van Slooten et al., 2019), specifically linked to striatal D1 and D2 receptors (Jonkees & Colzato, 2016), which are in turn broadly related to both cognitive and emotional control (Ayano, 2016). Prior work finds that eye blink rate and dopamine activity are positively related, where increases in eye blink rate are associated with increases in dopamine binding (Jonkees & Colzato, 2016; Karson, 1983; Van Slooten et al., 2019), although the exact mechanism underlying this association remains unclear (Bacher & Smotherman, 2004a). The eye blink to dopamine association has been validated through pharmacological studies in both animal (Groman et al., 2014; Karson, 1983; Kleven & Koek, 1996) and human (Jongkees & Colzato, 2016; Semlitsch et al., 1993) models, using dopamine agonists and antagonists, as well as in patient populations such as individuals with Parkinson’s disease (Fitzpatrick et al., 2011; Hall, 1945; Karson et al., 1982) or Schizophrenia (Chan et al., 2010; Jongkees & Colzato, 2016). We do note, however, that research associating eye blink rate with striatal dopamine synthesis has been mixed (van den Bosch et al., 2023) and to our best knowledge no work has tested the association between dopaminergic activity and eye blink rate in healthy children, in part due to ethical and methodological considerations.” 

We have also added the following text to the discussion (pg 29-30), “Finally, we recognize that direct associations between eye blink rate and dopamine have emerged from a predominantly clinical literature (e.g., Chan et al., 2010; Fitzpatrick et al., 2011; Karson et al., 1982; Shook et al., 2005) as well as more invasive adult tasks (e.g., Badgaiyan, 2014; Borwick et al., 2020; Cervenka et al., 2012; Fukai et al., 2019). As such, limited work has examined associations between dopamine and eye blink rate in healthy children. Therefore, it may be the case that changes in eye blink rate are related to attentional processes more broadly (Bacher, 2014; Bacher & Allen, 2009; Bacher et al., 2017; Rac-Lubashevsky et al., 2017; Ranti et al., 2020; Siegle et al., 2008) and more distally related to dopaminergic activity itself.”

2. Could you provide proofs from the literature of the link between EBR and dopamine in healthy children? In absence of evidence, and this is my main concern about the conclusions of this study, spontaneous eye blink rate of healthy children can’t be used as a proxy of dopamine synthesis in absence of scientific evidence proving the relation between dopamine neurotransmission and EBR in this specific population. The discussion about dopamine should then be more nuanced. The last sentence of the conclusion should be revised. The use of “dopamine” as a key word for this study is then questionable.

Thank you for highlighting this concern in the manuscript. As in our response to Point 1, we have qualified the available data in the literature and added additional context. We believe that the literature reviewed, behavioral patterns noted, and our explicit qualifications provide a good foundation for our novel examination of blink rate in a child population.

We did revise our key words to “attention” rather than “dopamine” to reflect the reviewer’s suggestion. 

Minor issues:

3. In the introduction, authors should define more the terms “externalizing problems” and “internalizing problems”.

Thank you for this suggestion, we have revised as follows (page 3), “High levels of effortful control are frequently associated with behaviors tied to adaptive socioemotional development, including better emotion regulation (Kochanska et al., 2000), greater prosociality (Pereira et al., 2021), and reduced risk of externalizing problems such as ADHD and impulsivity (Achenbach et al., 2016; Eisenberg et al., 2009; Kim-Spoon et al., 2019; Pereira et al., 2021; Valiente et al., 2003). Effortful control is also frequently regarded as a protective mechanism against internalizing problems including anxiety and depression (Achenbach et al., 2016; Eisenberg et al., 2009; Kim-Spoon et al., 2019).”

4. In introduction 2nd paragraph, “additionally, the authors found … behaviours”: is this sentence a result of a study (then a quote is needed), or more a general observation/opinion?

Thank you for noting the omission. This text referred to a finding from Murray & Kochanska (2002) and we now have an additional citation for clarity. 

5. In Methods, concerning code of blink: the threshold used to determine between blink and sustained eye closure should be indicated.

We added to page 20 that sustained eye closure was defined as the eyes being closed for more than 1 frame. 

6. First sentence of “descriptive statistics” “With the relative novelty of this experimental design, I first sought to describe these behavioral and physiological measures within our sample and how they may correlate.” should be revised.

We thank the reviewer for identifying this error, it has been revised to, “With the relative novelty of this experimental design, we first sought to describe these behavioral and physiological measures within our sample and how they may correlate.”

7. The absence of baseline could be problematic, as you mention in discussion. In the trials 1 and 5, what was the mean duration of task execution? Is the EBR during those short periods an available data?

Thank you for this question. These short periods included no wait time for the child (the experimenter just took their turn, with no standardized timing) so we were not comfortable with using this as baseline data given the lack of standardized control. 

8. Concerning results, they should be more detailed in the text part. 

We have now noted more information in the text, supplementing the tables and figures. 

9. On table 4, what the 1 to 8 stand for?

1 through 9 aligns with the 9 variables on the y-axis of the correlation table, so as to save the space of supplying the full variable name on the y-axis. We have now added this note to the table description.

10. In discussion, could you comment on the fact that with successive trials, the game itself may become more difficult with maybe a more unstable tower and more at risk of falling, that may influence the attention onto the game by itself.

We thank the reviewer for this point. We have added the following to page 29, “Another explanation is that as the trials get more taxing, both in terms of wait time and the instability of the tower/the risk of the tower falling, task-related decreases in eye blink rate may be reflecting increases in sustained attention for these children (Bacher et al., 2013; Bacher & Allen, 2009; Ranti et al., 2020).”

11. In discussion, the sentence “Looking to changes in EBR within a task, work generally finds that increases in task demands relate to increases in task demands” needs to be rephrased.

Thank you for identifying this error, we have revised as follows, “Looking to changes in eye blink rate within a task, the literature generally finds that increases in task demands relate to increases in eye blink rate” (page 28).

---

## [Decision Letter · Decision Letter 1]

13 Nov 2023

Now it’s your turn!: Eye blink rate in a Jenga task modulated by interaction of task wait times, effortful control, and internalizing behaviors

PONE-D-22-35097R1

Dear Dr. Gunther,

We’re pleased to inform you that your manuscript has been judged scientifically suitable for publication and will be formally accepted for publication once it meets all outstanding technical requirements.

Kind regards,

Vilfredo De Pascalis

Academic Editor

PLOS ONE

Additional Editor Comments (optional):

The reviewer is glad of the revised manuscript, and I agree that the authors have addressed adequately all the suggested changes.

Considering that the revision process of the current study has required a time too long and the quality of the paper is entirely improved, I have decided to accept it.

I would like to thank the authors for their patience.

Reviewers' comments:

Reviewer's Responses to Questions

**Comments to the Author**

1. If the authors have adequately addressed your comments raised in a previous round of review and you feel that this manuscript is now acceptable for publication, you may indicate that here to bypass the “Comments to the Author” section, enter your conflict of interest statement in the “Confidential to Editor” section, and submit your "Accept" recommendation.

Reviewer #2: All comments have been addressed

2. Is the manuscript technically sound, and do the data support the conclusions?

Reviewer #2: Yes

3. Has the statistical analysis been performed appropriately and rigorously? 

Reviewer #2: Yes

4. Have the authors made all data underlying the findings in their manuscript fully available?

Reviewer #2: Yes

5. Is the manuscript presented in an intelligible fashion and written in standard English?

Reviewer #2: Yes

6. Review Comments to the Author

Reviewer #2: My previous comments have been adequately addressed.

I appreciate the nuance that has been made about blink rate as a proxy of dopamine activity, even though it could have been more nuanced, to my opinion, in the last paragraph of the conclusion.

However, I consider this article suitable for publication as it is now.

7. PLOS authors have the option to publish the peer review history of their article (what does this mean?). If published, this will include your full peer review and any attached files.

Reviewer #2: **Yes: **Quentin Salardaine

---

## [Editor Report · Acceptance letter]

21 Nov 2023

PONE-D-22-35097R1 

Now it’s your turn!: Eye blink rate in a Jenga task modulated by interaction of task wait times, effortful control, and internalizing behaviors 

Dear Dr. Gunther:

I'm pleased to inform you that your manuscript has been deemed suitable for publication in PLOS ONE. Congratulations! Your manuscript is now with our production department. 

Kind regards, 

on behalf of

Prof. Vilfredo De Pascalis 

Academic Editor

PLOS ONE